# Exhaled Carbon Monoxide Level and Practices among Tobacco and Nicotine Adult Users in Klang Valley, Malaysia

**DOI:** 10.3390/ijerph20054443

**Published:** 2023-03-02

**Authors:** Sharifa Ezat Wan Puteh, Norayuni Mohd Ismail, Zaleha Md Isa, Andrea Yu-Lin Ban

**Affiliations:** 1Department of Community Health, Faculty of Medicine, Universiti Kebangsaan Malaysia, Jalan Yaakob Latif, Cheras, Kuala Lumpur 56000, Malaysia; 2Respiratory Unit, Faculty of Medicine, Universiti Kebangsaan Malaysia, Jalan Yaakob Latif, Cheras, Kuala Lumpur 56000, Malaysia

**Keywords:** cigarette, electronic cigarette, heated tobacco product, exhaled carbon monoxide, poly-users, switching

## Abstract

Tobacco and nicotine derivatives uses are multiple in nature. These include conventional cigarettes (CCs), heated tobacco products (HTPs), and electronic cigarettes (ECs). This study aims to determine the practices, nicotine dependency profile, association with exhaled carbon monoxide (eCO) level, and pulmonary function (PF) among adult product users and non-smokers. This cross-sectional study involved smokers, nicotine users, and non-smokers from two public health facilities in Kuala Lumpur from December 2021 to April 2022. Data on socio-demography, smoking profile, nicotine dependency level, anthropometry, eCO monitor, and spirometer measurements were recorded. Out of 657 respondents, 52.1% were non-smokers, 48.3% were CC only smokers, poly-users (PUs) (27.3%), EC-only users (20.9%), and HTP-only users (3.5%). EC use was prevalent among the younger aged, tertiary educated, and females; HTP use was prevalent among those of an older age and CC users was common among lower educated males. The highest median eCO (in ppm) seen were as follows: in CC users only (13.00), PUs (7.00), EC users (2.00), HTP users (2.00), and the least was observed among non-smokers (1.00), which is significantly different across the groups (*p* < 0.001). Comparison of practice between the different product users showed significant differences in age of product initiation (*p* < 0.001, youngest in CC users in PUs), duration of product use (*p* < 0.001, longest in exclusive CC users), cost per month (*p* < 0.001, highest in exclusive HTP users) and attempt to quit product (*p* < 0.001, CC use in PUs had the highest attempt to quit), while there is no significant difference in Fagerström score across the groups. Among EC users, 68.2% successfully switched from smoking CCs to ECs. The findings suggest that EC and HTP users are exhaling less CO. The use of these products in a targeted approach may manage nicotine addiction. Switching practice was higher among current EC users (from using CCs), hence emphasizing the need of switching encouragement and total nicotine abstinence later on. Lower eCO levels in the PU group, (as compared to CC-only users) and high quit attempt rate among in CC use in PUs may indicate attempt of PUs in reducing CC use through alternative modalities such as ECs and HTPs.

## 1. Introduction

It is known that there is no safe level of exposure to tobacco. Whilst all forms of tobacco are harmful, conventional cigarette (CC) smoking is the most common form of tobacco consumption worldwide [1]. A CC is a narrow cylinder containing materials, which is typically made of finely cut, cured tobacco leaves and is wrapped in thin paper. It may also contain other ingredients, including substances that can add different flavors. A cigarette is ignited on one end and inhaled into the lungs [2]. Apart from that, there are novel and emerging nicotine and tobacco products such as heated tobacco products (HTPs) and electronic cigarettes, commonly referred to as e-cigarettes (ECs). HTPs function by heating the tobacco just enough to release a nicotine-containing tobacco vapor, but without tobacco combustion. This product is not intended as a method of CC cessation but has been promoted as a means of harm reduction products for smokers [3]. Meanwhile, ECs, which are battery powered electronic devices that are known as electronic nicotine delivery systems (ENDS) and electronic non-nicotine delivery systems (ENNDS) and function by heating a solution, or they are known as e-liquids, which typically consist of propylene glycol or glycerol (glycerin), nicotine, and flavoring agents [4] to create a vapor that condenses to form an aerosol [1].

Although admirable progress is being made globally in terms of the reduction in tobacco use as evidenced by the declining rates of tobacco use prevalence, the battle of the tobacco epidemic is far from over. One of the nine voluntary global targets for the prevention and control of the non-communicable disease (NCD) target under the World Health Organization (WHO) Global Action Plan for the Prevention and Control of NCDs 2013 to 2020 focuses on behavioral risk factors, one of which is tobacco use. Global tobacco use has shown a downtrend pattern. It was reported that in 2000, an estimated 32.7% of the global population aged 15 years and older were current users of some form of tobacco, whereas in 2020, the prevalence declined by one third to 22.3%.

Nevertheless, to date, global estimate data from the WHO on EC use are not available yet due to the lack of population-based survey data from many more countries. In Great Britain, representative surveys reported that EC use prevalence among adults has plateaued since 2015. Estimates for prevalence were between 5.4% and 6.2% for all adults in 2017 to 2018 [5]. While a study on ECs that involved seven countries, namely, Russia, Poland, Germany, UK, France, Italy, and South Korea, described that the usage of ECs increased by up to 85% from 2013 to 2015, which indicates an increase from 2.8 million to 5.1 million [6]. From the available surveys with the age ranges as more than 15 to 20 years old, the highest prevalence of EC use was reported as 8% among persons aged 18 years and older in Iceland in the 2017. Moreover, global data specifically on HTP use is not available at this moment as it is summed up cumulatively in smoked tobacco data [7].

At a local level, the Malaysia National Health Morbidity Survey (NHMS) reported a slight reduction in the prevalence of current smokers, from 22.8% in 2015 to 21.3% in 2019. For respondents who currently smoked CCs, the overall prevalence was 20.8%, which encompassed 39.6% male respondents and 1.2% female respondents. Meanwhile, the prevalence of ECs was reported as 4.9% and the use was highest among those in the 20 to 24 years age group [8]. However, there are no specific data on HTPs prevalence in the local context.

Almost half a million adults will still die prematurely because of tobacco smoking. Annually, the total economic costs due to tobacco are now over USD 289 billion and it is projected that 5.6 million children alive today who are younger than 18 years of age will die prematurely as a result of smoking. Health effects of smoking toward smokers has always been known to cause diseases such as Chronic Obstructive Pulmonary Disease (COPD), lung cancer, cardiovascular disease, and Type 2 Diabetes Mellitus as well as causing systemic inflammation and impaired immune function [9].

Furthermore, there are many concerns over the impact of the use of ECs toward health. It is important to realize that EC emissions are not merely “harmless water vapor” [4]. Unadulterated ECs produce aerosols that contain various substances such as glycols, volatile organic compounds, and metals. Although on average, it has been reported that the level is much lower than that of CCs, it can still be higher depending on the type of device, as in an open system device, the temperature can be increased, which can cause an increased thermal decomposition of the e-liquid [10] Other than those substances, aerosols contain nicotine in ENDS. Nicotine is a well-known substance that can cause various health effects, for instance cardiovascular diseases, and it is known to affect brain development [10]. ECs also can expose users to high levels of ultrafine particles that pose cardiovascular disease risks and noncancerous lung disease risks, which are similar to the effects of exposure to CCs. Apart from that, it can be a source of indoor air pollution [4] and second-hand aerosol, which is a passive exposure to emissions from EC users [10]. Furthermore, although there is much research that focuses on the prevalence and factors associated with the use of ECs, there is scarce evidence that associates the practice of EC use with clinical outcomes, particularly in a Malaysian setting.

Socio-economic factors have been showed to affect smoking pattern. A study in Korea showed that the lowest household income group had a higher risk of smoking than the highest household income group in both urban and rural areas [11]. Another study reported that greater nicotine dependence and those who reported smoking more cigarettes per day were more likely to report both difficulties living within their household’s income restraints and a greater concern about food and housing expenses [12]. In the Malaysian socio-economic setting, there are three different population groups according to the income levels for each household, namely, the T20, M40, and B40 groups. T20 represents the 20% highest earners in Malaysia; M40 represents the 40% middle earners, and B40 represents the lowest 40% earners [13]. A study conducted in Malaysia reported that Human Development Indicators (HDIs) are inversely related to smoking consumption, which means that when the life expectancy, education level, and income per capita start to increase, cigarette consumption begins to decrease and vice versa [14].

Hence, efforts to reduce the burden of diseases attributed to tobacco smoking has been accepted and implemented to stop the addictiveness of nicotine. One of the widely recognized efforts is via the use of nicotine replacement therapy (NRT) [15]. However, according to a study, although long-term nicotine replacement therapy provides an option for smokers, it does not result in greater rates of cessation or harm reduction [16].

Furthermore, with the emergence of poly-tobacco users, various switching practices, the rapid evolution of ECs and contraband cigarettes in the market, and also the substantial health complications of tobacco and nicotine derivative products to the community, there is a vital need to explore these subjects. Thus, this study aims to determine the practices, nicotine dependency profile, and association between exhaled carbon monoxide (eCO) level and pulmonary function among adult users and non-smokers. This article will describe the practices, nicotine dependency profile, and association with exhaled carbon monoxide (eCO) levels among adult CC, EC, and HTP users and non-smokers. The pulmonary function outcome in this study will be described in another article.

## 2. Materials and Methods

A cross-sectional study was performed to determine the practices, nicotine dependency profile of smoking variations, and its association with pulmonary function and eCO level among adult EC, HTP, and CC users and non-users. Total study duration was 12 months, which started on July 2021 and ended in August 2022. Pilot study was started on the first week of December 2021 and subsequently, the cross-sectional study was commenced daily until April 2022. May to June 2022 was for data cleaning and data analysis. Sample population was Malaysian population aged 18 to 60 years old who attended the outpatient clinics, the caregivers accompanying the patients attending the outpatient clinics under a tertiary education center in Kuala Lumpur, Malaysia, and also residents living in the surrounding area. The inclusion criteria were Malaysian citizens, adult aged 18 to 60 years old, single CC users, EC users, HTP users, poly-users (PUs), or non-users of any of the products mentioned. For users of CCs, ECs, or HTPs, they must have used it for at least six months and they must be able to communicate and read in either Malay language or English. The exclusion criteria were people who have medical or surgical contraindications for spirometry test, those who have been diagnosed with chronic respiratory diseases such as COPD and asthma, and those who have been diagnosed with COVID-19 infection and pregnancy. The sample size was calculated using a web-based calculator, OpenEpi (Version 3) [17], which calculates results by comparing two independent means. In obtaining the total sample size, the design effect of five arms, which was planned in this study, was considered. With the assumption of dropout rate of 20%, the total sample size was 732. A non-probability sampling technique was used as the sampling method for this study, which was convenience sampling based on voluntary basis. This method was chosen because members of the target population met certain practical criteria, such as easy access to the study sites, geographical proximity, availability at certain time, and the willingness to participate for the purpose of the study. Due to uncertainties of the COVID-19 pandemic at the time of the data collection, this sampling method was deemed appropriate and feasible. There were several instruments used in this study. First is a questionnaire that was adopted and adapted based on the available literature; second is the devices measuring blood pressure and anthropometric indices, which included blood pressure reading, weight, and height scale. Thirdly, a carbon monoxide analyzer, and finally, the spirometer.

The questionnaire used in this study was adopted from a combination of three validated questionnaires and it consisted of four sections, namely, the background of the respondents, current use of CC and Fargerström Nicotine Dependency test, current use of EC products and Penn Index, and current use of HTP products and Fargerström Nicotine Dependency test. Participants were only required to answer sections of the questionnaire based on their current use. The questionnaire was designed as such so that if a respondent did not consume a certain substance, the respondent could automatically skip the particular section. For non-smoker participants, they were required to answer only section A of the questionnaire (socio-demographic). The blood pressure and pulse rate of the respondents were taken using a fully automated blood pressure monitor by the operator. For weight and height measurement, the operator used a weighing scale and stadiometer, which were calibrated in both study sites. Measurements were documented by the operator accordingly. All measurements were performed once; however, for abnormal readings of blood pressure, the measurement was repeated after 15 min.

The exhaled carbon monoxide (eCO) levels were measured using a hand-held carbon monoxide monitor, which was the picO™ Smokerlyzer^®^. The measured eCO levels were based on the manufacturer’s recommendations and were standardized throughout the study. Participants were asked to inhale deeply, hold their breath for 15 s, and then slowly but forcefully blow into the disposable mouth piece, with the aim to empty the lungs completely. The procedure is conducted by one trained personnel in a well-ventilated room. Measurement of eCO was recorded in a pre-formatted form, which was the line listing for the day for all respondents, while the spirometry findings were kept in the spirometry software and were downloaded in an excel sheet. The questionnaire and clinical measurements were then combined into one excel form in Google Drive, which was password protected. Those with abnormal lung function tests detected during the study were offered referral to pulmonologist in Respiratory Unit, HCTM, for further intervention. Those who were interested in quitting smoking were also referred to the HCTM Quit Smoking Clinic. Subsequently, the data were transferred into the Statistical Package for Social Science (SPSS) version 22.0 for data analysis. Descriptive and inferential analysis were conducted with *p* ≤ 0.05 as significant. Approval for this study was obtained from the Universiti Kebangsaan, Malaysia, Faculty of Medicine Ethics Committee (JEP-2020-684). This study was funded by the fundamental grant under Faculty of Medicine, Universiti Kebangsaan, Malaysia, and the Malaysian Society of Harm Reduction.

The results of this study discussed in this article consist of socio-demography, comparison between groups, eCO levels, and switching practices. Results on pulmonary function are described and discussed in another article.

## 3. Results

### 3.1. Socio-Demographic and Smoking Status

Descriptive analysis of the socio-demographic profile of the study population is shown in Table 1. Categorical data are presented as the frequency and percentage, while continuous data were presented as the mean and standard deviation (SD). Participants’ age ranged from 18 to 60 years old, with a mean value age of 34.4 (SD ± 0.37). According to the age category, the 31–60 years age group is the highest proportion (61.9%) and the majority of the study population are male (70.3%). Meanwhile, according to ethnicity, Malay people made up the highest proportion (87.8%), followed by Indian people (5.9%), and Chinese people (4.1%). In terms of education level, the majority attained secondary education (53.0%) followed by tertiary education (43.4%). The majority of the participants are from the government sector (49.6%), and most of the participants are from the B40 income level (73.4%).

Descriptive analysis of the smoking status of participants is described in Figure 1. The current non-smokers are slightly higher among the respondents (52.1%) and among the current non-smokers, 80.7% had never smoked before, while 19.3% were former smokers. Meanwhile, among the tobacco smokers and EC users (47.9%), the majority are CC users (48.3%), followed by PUs (27.3%), then EC users (20.9%), and HTP users (3.5%). PUs were further subdivided into dual users (91.9%) and triple users (8.1%), with the use of CCs and ECs in dual users, and the use of CCs, ECs, and HTPs in triple users.

The demographics of participants according to their smoking status is described in Table 1. The age categories are divided into 18–30 and 31–60 years old. In both age categories, the majority of participants are non-smokers. Meanwhile, among the users, the CCs group is the most prevalent, which amounts to 19.2% among 18–30 year olds and 25.6% among 31–60 year olds. The ECs group is more prevalent among the younger age group (15.6%) compared to the older age group (6.6%). Conversely, in the HTPs group, the prevalence is higher among the older age group category (2.5%) as compared to the younger age group (0.4%). Overall, the mean age showed significant age differences across the different groups, with heated tobacco users recording the highest mean age of 39.18 (±7.39), and EC users recording the lowest mean age of 30.32 (±7.36).

According to gender, non-smokers was more prevalent among female (86.7%) compared to male (37.4%). Among male uses, CC use has the highest prevalence (31.4%), followed by poly-users (17.6%), EC users (11.2%), and then HTP users (1.9%). Among the female users however, the highest prevalence is among EC users (7.3%), followed by CC users (3.1%), then poly-users (2.1%), and HTP users (1.0%). In terms of ethnicity, it is divided into Malays and non-Malays. The latter consists of Chinese, Indian, Sikh, Bumiputera Sabah, and Sarawak ethnics. Use of CCs is highest in both Malay (23.9%) and non-Malay (17.5%). However, the second most prevalent group among Malays was the PUs (13.7%), meanwhile, among non-Malays, the second most prevalent group was HTP users (10.0%).

The prevalence of smoking in terms of marital status showed the same pattern among married and non-married groups: the highest is CC users, followed by poly-users, EC users, and HTP users. According to education level, the prevalence of smoking patterns is different. The highest prevalence among respondents with a tertiary education are EC users (11.9%), followed by poly-users (10.5%,) and then CC users (8.4%). Nevertheless, among the non-tertiary education respondents, the higher prevalence is for CC users (34.4%), followed by EC users (14.0%), and then poly-users (9.7%).

Furthermore, there was a similar smoking pattern among the respondents in all occupation categories, whereby the highest prevalence was for CC users, followed by PUs, and then EC users. Interestingly, HTPs are used more among private sector respondents and the non-B40 income earners. The latter finding is similar to a study in Japan that showed that a high income earner has higher odds of using HTPs [18] which is likely related to higher purchasing power. Meanwhile, CC users, EC users, and PUs are more prevalent among the B40 income group followed by PUs and the EC group. Furthermore, it is observed that the co-morbidity prevalence is higher among the non-smokers and is exclusive to the CC users. This could be due to the fact that a large proportion of participants were among the B40 income earners group and because researchers have shown that income level has an association to health status [19,20,21].

### 3.2. Comparison of Practices and Nicotine Dependency Profile

The overall comparisons of practices and nicotine dependency profiles between groups, as described in Table 2, were conducted in comparing five aspects, namely, age of product initiation, duration of product use in years cost per month (in MYR), Fagerström score, and any attempt to quit those products used. In terms of the initial age of the product used, CC only users and PUs indicated the earliest age with a mean age of 17.41 (±3.84) and 17.16 (±3.66) years old, respectively. This was followed by EC-only users and ECs and HTPs users in PUs at a later age of late 20s. Meanwhile, HTP-only users are seen to start at the higher mean age of 35.00 (±7.47) years old, where the user’s income is considered more stable. These findings were significantly different between groups (*p* < 0.001). The age of product initiation is most likely due to previous exposure to CCs that are available in the market much longer, as compared to other products and HTPs is more common in the older age group due to a greater purchasing power. The duration of product use is similar to the age trend of product initiation, whereby CC-only users and CC use in the PU group indicated a longer duration of product use at 17.58 (±9.74) and 13.93 (±7.76) years. The HTP users in the PU group showed the shortest mean duration of HTP use at 3.43 (±1.51) years. There are significant differences between the groups (*p* < 0.001). The monthly cost of the products, including its consumables, showed significant differences between groups (*p* < 0.001), whereby EC-only users spent the least cost per month at the mean value of MYR 81.86 (±96.30) and MYR 82.49 (±97.86) in EC users in the PU group. The highest monthly cost incurred was among the HTP-only users, with the mean value of MYR 330.64 (±195.56) per month. This is most probably due to the costly consumables of HTPs as compared to the cheaper refillable EC juices. Nicotine dependency, measured through the Fagerström score, showed no significant mean score difference between the groups (*p* = 0.109). The highest mean score was in the EC-only group at 4.12 (±2.39), and the lowest was among HTP-only group at 2.55 (±2.21).

The attempt to quit product use was significantly different among the groups (*p* < 0.001). It was most prevalent among the in PU group who used CCs (86.0%) and was the lowest among the HTP-only users (9.1%). This may indicate the higher intention of CC users in the PU group to switch to ‘safer’ products.

### 3.3. Switching Patterns

There are various patterns of product use that can be seen in this study among the single product users (CC, EC, and HTP) and poly-users, as depicted in Figure 2.

#### 3.3.1. Single Product User

Out of the 152 CC smokers, 52 (34.2%) CC smokers have used ECs in their lifetime. Among these 52 smokers, there are three patterns of use. Firstly, 48 people (92.3%) of this group started with CCs, subsequently added ECs, and then quit ECs and reverted to smoking CCs again. Three smokers (5.8%) of this group started CCs and ECs in the same year (dual user) and then quit ECs and continued smoking CCs. Only one person (1.9%) in this group initiated smokeless tobacco with ECs in the previous year, then quit ECs, and then smoked CCs. For the EC-only group, out of 66 users, 45 (68.2%) EC users have used CCs previously. Among the 45 EC users, there are two patterns of use. Firstly, 41 people (91.1%) of this group switched from CCs to ECs. Four people (8.9%) started CCs and ECs in the same year, then stopped CCs and thereafter continued ECs only. For the HTP-only group, eight out of eleven people (72.7%) have used more than one product. The two usage patterns showed that they started with CCs. Firstly, five people (62.5%) started with CCs and then added ECs, and subsequently started with HTPs and stopped both CCs and ECs. Meanwhile, three people (37.5%) from this group switched from CCs to HTPs.

#### 3.3.2. Poly-User Group

The poly-user group consisted of 86 people whereby 79 (91.9%) of them were dual users and 7 (8.1%) were triple users. A total of 74 people (93.6%) among the dual users started with CCs, then subsequently added ECs to their use. Secondly, four people (5.1%) started CCs and ECs in the same year and one person (1.3%) started with ECs and subsequently added CCs. Further analysis showed that in the pattern there was a significant reduction in the average CC use from 17.87 (±13.16) sticks per day to 11.26 (±9.41) sticks per day after adding EC use (*p* < 0.001), whereby 70.0% reduced the quantity of CCs, while 30% did not changed the quantity of CCs despite adding ECs. However, none increased their CC quantity of sticks per day after adding EC use.

For triple users, three (42.8%) started CCs first and then subsequently added ECs, and then later added the use of HTPs. Meanwhile, another pattern showed that one person (14.3%) started off with CCs and then subsequently started with HTPs and ECs thereafter. The third pattern showed that two people (28.6%) initiated with CC use then started ECs and HTPs simultaneously in the same year. Additionally, one person (14.3%) started off with CCs and ECs in the same year, and then subsequently added HTPs. All showed a reduction in CCs smoked prior to being a triple product user.

### 3.4. Comparison of Exhaled Carbon Monoxide Level between Groups

Table 3 showed the mean exhaled carbon monoxide level according to the products used, and a comparison was made between product users and the non-smokers. The highest median exhaled CO level was among the CC users, followed by PUs, EC users, and then HTP users. Current non-smokers exhibited the lowest median eCO level. There was a significant difference in the median eCO levels across the groups (*p* < 0.001). Poly-users reduced their CC use and switched to ECs or HTPs, which contributed to the lower eCO values.

The eCO level categories according to different products users and non-smokers are depicted in Table 4. All of the HTP users were within the 0–6 ppm CO category. The CC-only users presented the highest values of more than 10 ppm CO exhaled (66.4%), which showed significant differences across the groups (*p* < 0.001). Pearson Chi-square test indicates that there was a significant difference between the different groups in terms of the eCO levels.

## 4. Discussion

According to the global data reported in 2020, the prevalence of overall and male cigarette smoking is lower than our findings, at 15.5% and 26.4% [7]. This may reflect the true picture as described in the WHO data, whereby the proportion of cigarette smokers is highest in the Western Pacific Region, where Malaysia is located [7]. Nevertheless, among females, the prevalence of cigarette smoking in our study is lower as compared to global level, at 4.6% in 2020 [6]. Furthermore, the overall prevalence of female CC-only users in our study is higher as compared to a national representative survey conducted in Malaysia in terms of the overall female prevalence of cigarette smoking, at 20.8% and 1.2%, respectively. However, CC smokers among males showed a higher prevalence in the nationwide study at 39.6% compared to what we reported [8].

Additionally, the prevalence of tobacco users in our study is higher as compared to another local study, which reported a prevalence of tobacco use in rural and urban areas at 25.4% and 20.1%, respectively [8]. This may be due to the nature of our study sampling, which uses convenience sampling, also being contributed by the timing of the study during the pandemic of COVID-19. There are studies that have showed that as the pandemic and lockdown progressed, patients smoked more cigarettes, which would indicate that the COVID-19 pandemic significantly contributed to an increase in smokers [22]. Moreover, another study showed that 45.8% of the participants increased their smoking during the COVID-19 lockdown. The odds of smoking more during the lockdown were associated with a higher subjective stress due to the COVID-19 pandemic [23].

Our study also showed that CC-only users are more prevalent in the 31–60 year old age group, non-tertiary educated group, and those who are in the private sectors. These findings were somewhat consistent with the 2019 NHMS report, whereby the prevalence of current smokers peaked in the 30–34 years age group, with tertiary education attainment having the lowest prevalence, and the highest prevalence of current smokers was among those who were self-employed followed by private sector employees [8]. Another study conducted in China also reported that CC users were more common among those who are older, male, with a lower level of education, and having been exposed to second-hand smoke [24], which is similar to our study findings with the exception of the second-hand smoking status. The prevalence of exclusive CC use among users in this study (48.3%) is similar to another study among young adults (46.5%), but that study included other tobacco products including cigars and hookah, which were not included in our study [25].

Furthermore, the prevalence of EC-only users in this study is 10.0%, which is higher than the national survey conducted in Malaysia. Its findings reported that the prevalence at the national level and the Federal Territory of Kuala Lumpur is at 4.9% and 3.1%, respectively. The highest prevalence of EC use was reported in the Federal Territory Labuan at 8.2% [8]. The non-randomized nature of the sampling of this study may contribute to these findings. However, so far, there are no representative global data on the current use of ECs to compare to, as population-based survey data on EC use are not yet available from enough countries to obtain a global estimate of EC use and its prevalence among adults [7]. Similar to our findings, Zhu et al. also reported that EC users were predominantly male and of a younger age group [24].

Moreover, our study reported that the PU group was more prevalent among the younger age groups, which showed mixed findings in several studies [26,27]. Similarly, PUs patterns differ across socio-demographic factors, including gender and ethnicity [27,28]. PUs are also more prevalent compared to exclusive EC users (27.3% vs. 20.9%), which is consistent with another study; however, the dual users were five times more common than exclusive EC users [26]. The marked difference of prevalence between the two groups may be due to the study population, which are primarily adolescents. A Korean study reported that dual group users were more common in younger age groups, those with high education attainment, those with a higher income level, those living in urban areas, and professional workers compared to CC-only smoker. Dual users were also associated with higher psychosocial, behavioral, and cardiovascular risk factors when compared to CC-only users [29].

HTP users in this study are more prevalent among the non-B40 group and the older age group. The former is also consistent with a study conducted in Japan whereby more affluent people tended to use HTPs [18]. One possible reason for this finding is due to the higher prices of HTPs compared to combustible cigarettes [30]. However, in our study, HTPs are seen more in the older age group, which contrasts with a study reported in Japan, where younger people were more likely to use HTPs [18].

In comparing nicotine dependency, this study showed that EC users in the single and PU group had higher Fagerström scores as compared to CC users in the single and PU group; however, no significant differences were seen. This is similar to another piece of research that reported nicotine dependence levels to be higher among EC-only users and dual users compared to traditional tobacco smokers [31]. These findings suggest that ECs may have a higher addictive potential than smoked tobacco cigarettes among adults. Additionally, a study in South Korea reported that poly-tobacco product users reported higher nicotine dependence symptoms than single users [32]. This finding contrasts to our study, whereby the PU group has a lower nicotine dependence as compared to single users of CCs and ECs, with the exception of HTP-only users. This may indicate the intention of PUs to switch toward other products or the efforts to stop smoking. Furthermore, one study showed that PUs were also associated with increased attempts to quit CCs [33] and this is consistent with our findings, whereby CC users in PU group have a higher prevalence of attempting to quit as compared to single CC users (86.0% vs. 55.9%). Moreover, a study in Korea reported that exclusive HTP and EC users were approximately 40% and 20% less likely to quit the product that they used than exclusive CC smokers, respectively [34]. This is similar to our findings, whereby attempts to quit in single product users is the highest among CC users (55.9%), followed by EC users (34.8%), and then HTP users (9.1%). This observed trend is similar in the PU group. Consistently, these results suggest the possibility that people who use HTPs have less interest in attempting to quit tobacco and that people using CCs have a higher interest in attempting to quit smoking. However, given the small sample size of HTP users in this study, this result has to be interpreted with cautious.

In terms of switching, our study reported that in the current exclusive CC group, only one person among this group (0.7%) started with ECs and end up switched to CC only. This very low switching prevalence indicates that ECs may not be the main contributor to the initiation of cigarette smoking. Nevertheless, this is contrary to a study by Berry et al. that reported that 21.8% of new cigarette users (178,850 youths) and 15.3% of current cigarette users (43,446 youths) among U.S. youths may be attributable to prior EC use, which support the notion that the use of ECs may be a contributor to the initiation of CC smoking among youths [35].

In the exclusive EC group, 68.2% of them switched from CC use to exclusive EC use. One qualitative study explored this matter and reported that the interest in trying ECs relates to the hopes of quitting CCs and due to convenience. Moreover, satisfaction, lack of product problems, and perceived safety facilitated the successful switching from CCs to exclusive EC use. However, they reported trying many products before they found ones that satisfied their needs [36], which is not explored in our study. A different study indicated a great potential for an increase in the frequency of relapse to CCs in those who switched to a regular use of ECs. It is reported that among former CC users who are current EC users, they have double the risk of relapse to CCs. Long-term former smokers were identified as the main contributing factor for relapse [37].

In the PU group, there was a significant reduction in the average CC use from 17.87 (±13.16) sticks per day to 11.26 (±9.41) sticks per day after adding EC use (*p* < 0.001), whereby a majority reduced the quantity of CCs and none increased the quantity pf their CC sticks per day after adding EC use. This significant finding indicates that most PUs are reducing their CCs use when initiating ECs. This is similar to a study that reported that participants significantly decreased their cigarette consumption per day from pre-to post-vaping; however, the study showed that the total nicotine use and dependence increased [38]. One study in the UK reported that relapse to smoking is likely to be more common among former smokers that vape infrequently or use less advanced devices [39]. Overall, this indicates the importance of regulating the use of ECs, as it can be a potential method of reducing the harm of CC use. For instance, in the UK, ECs have been acknowledged as a tool for assisting smoking cessation and a study involving a pharmacy supported EC smoking cessation intervention showed that EC distribution combined with pharmacy support appears to be an acceptable and effective intervention for smoking cessation [40].

The exhaled CO level is a method that can be used as an objective, non-invasive indicator to determine an individual’s smoking status and distinguish smokers from non-smokers [41,42,43]. It is also reported that eCO levels were significantly correlated with nicotine dependency [44]. Besides that, the severity of smoking can be quantified by the eCO and blood COHb% levels as these parameters correlate with the severity of smoking as the number of cigarettes per day, the duration of smoking, and the Brinkman index [45]—which is a measure of cumulative smoking exposure [46].

Our study demonstrated that CC-only users reported the highest average median eCO level, followed by PUs, EC users, HTP users, and then the non-smoker group. In contrast to our findings, a study in Korea on a different smoking biomarker, urine cotinine, reported the highest urine cotinine being among the PU group as compared to CC smokers, EC user, and the non-smoker group. In this case, the dual users have higher cotinine levels than the other groups, which may indicate that they take more nicotine via cigarettes or ECs, or are more addicted than others [47]. However, our study showed that eCO levels in the PU group were lower than the CC-only users, which may indicate that the PU group is more likely to use other products to reduce their CC use. This is correspondingly consistent with our findings while investigating the switching of products, whereby a significant reduction in CCs can be seen after the initiation of ECs in the PU group. ECs and HTPs reported the lowest eCO level among the other products used in this study, with an average median level of 2.0 ppm. It is known that HTPs are a weaker indoor pollution source than CC; nevertheless, their impacts are neither negligible nor fully understood [48]. It is noticeable that the median eCO level of CC-only users was thirteen times more than the non-smokers (13.0 vs. 1.0), which is much higher as compared to a study in China, whereby CC users reported only three times more average eCO level as compared to non-smokers [49]. Factors that may contribute to these differences include the number of cigarettes smoked, degree of inhalation, background level of CO from other sources, ethnical differences in alveolar ventilation, or the CO diffusion capacity of the lungs between various geographical populations [49]. Additionally, our findings are also consistent with another study that required participants to measure eCO levels at baseline, followed by series of measurements up to 45 min after the use of CCs, ECs, and HTPs. In that study, there was a significant increase in eCO levels for CCs as compared to ECs and HTPs. Peak eCO levels of CC users were significantly higher as compared to EC and HTP users [50], which is consistent with our findings.

Nevertheless, there are several limitations that should be considered when interpreting the results due to the nature of this study. Firstly, the participants in this study were recruited using non-probability sampling methods. Hence, it is not representative of the general population and therefore, other future representative studies are needed. Secondly, as this is a cross-sectional study, the causality as well as temporal relationships could not be concluded. Next, the possibility of recall bias is present as some of the data obtained were from self-reported questionnaires. Additionally, the small sample size in certain product users, particularly the HTP user group, is another limitation as the findings may not be generalizable.

## 5. Conclusions

The usage patterns of tobacco and nicotine derivative products can be seen in this study. CC use has the highest prevalence, followed by the PU group, EC users, and then HTP users, and the pattern differs according to socio-demographic factors. There are significant differences seen in the different product use in terms of age of starting the product, attempting to quit smoking, and also the cost of the product. Although there are differences in the mean Fagerström score among the groups, they are not significant. Exhaled CO levels between the different groups showed significant differences and were consistent with other available studies. Lower eCO levels in the PU group as compared to CC-only users may indicate the attempt of users in reducing CC consumption through alternative modalities such as ECs and HTPs.

## Figures and Tables

**Figure 1 ijerph-20-04443-f001:**
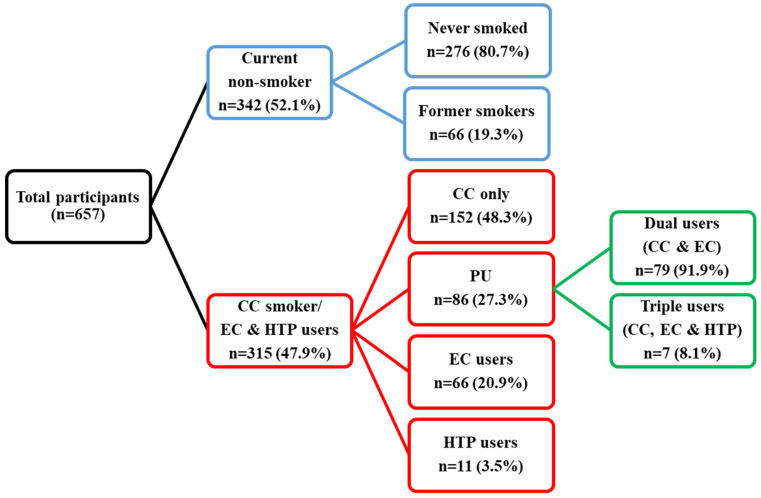
Description of current smoking status among participants.

**Figure 2 ijerph-20-04443-f002:**
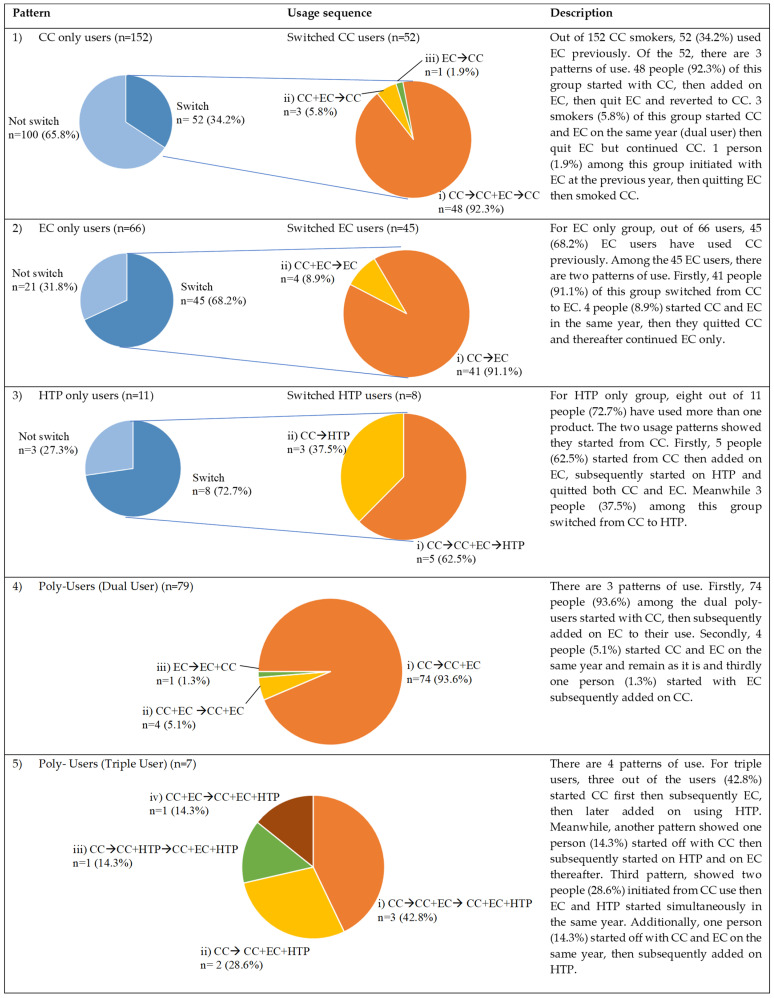
Pattern and usage sequence of product use.

**Table 1 ijerph-20-04443-t001:** Demographics of study population according to smoking status (n = 657).

Variables (n = 657)	Overall n (%)	Non-Smokers	CCs	PUs	ECs	HTPs	*p*-Value
		(n = 342) n (%)	(n = 152) n (%)	(n = 86) n (%)	(n = 66) n (%)	(n = 11) n (%)	
**Age (years)**							<0.001 *^a^ <0.001 *^b^
Mean (±SD)	34.4 (±0.37)	34.85 (±9.78)	36.20 (±9.47)	31.89 (±8.09)	30.32 (±7.36)	39.18 (±7.39)	
18–30	250 (38.1)	120 (48.0)	48 (19.2)	42 (16.8)	39 (15.6)	1 (0.4)	
31–60	407 (61.9)	222 (54.5)	104 (25.6)	44 (10.8)	27 (6.6)	10 (2.5)	
**Gender**							<0.001 *^b^
Male	462 (70.3)	173 (37.4)	146 (31.4)	82 (17.6)	52 (11.2)	9 (1.9)	
Female	195 (29.7)	169 (86.7)	6 (3.1)	4 (2.1)	14 (7.3)	2 (1.0)	
**Ethnicity**							<0.001 *^b^
Malay	577 (87.8)	298 (51.6)	138 (23.9)	79 (13.7)	59 (10.2)	3 (0.5)	
Others	80 (12.2)	44 (55.0)	14 (17.5)	7 (8.8)	7 (8.8)	8 (10.0)	
**Marital status**							0.047 *^b^
Married	407 (61.9)	210 (51.6)	108 (26.5)	45 (11.1)	38 (9.3)	6 (1.5)	
Non-married	250 (38.1)	132 (52.8)	44 (17.6)	41 (16.4)	28 (11.2)	5 (2.0)	
**Education level**							<0.001 *^b^
Tertiary	285 (43.4)	189 (66.3)	24 (8.4)	34 (11.9)	30 (10.5)	8 (2.8)	
Non-tertiary	372 (56.6)	153 (41.1)	128 (34.4)	52 (14.0)	36 (9.7)	3 (0.8)	
**Occupation**							<0.001 *^b^
Government staff	326 (49.6)	204 (62.6)	62 (19.0)	34 (10.4)	26 (8.0)	0 (0.0)	
Private sector	235 (35.8)	67 (28.5)	77 (32.8)	45 (19.1)	35 (14.9)	11 (4.7)	
Unemployed/Student	96 (14.6)	71 (74.0)	13 (13.5)	7 (7.3)	5 (5.2)	0 (0.0)	
**Income group**							<0.001 *^b^
B40	482 (73.4)	213 (44.2)	138 (28.6)	73 (15.1)	56 (11.6)	2 (0.4)	
Non B40	175 (26.6)	129 (73.7)	14 (8.0)	13 (7.4)	10 (5.7)	9 (5.1)	
**Co-morbidity**							<0.001 *^b^
Absent	507 (77.2)	251 (49.5)	113 (22.3)	72 (14.2)	61 (12.0)	10 (2.0)	
Present	150 (22.8)	91 (60.7)	39 (26.0)	14 (9.3)	5 (3.3)	1 (0.7)	
**Weight (kg)**							0.347 *^a^
Mean (±SD)	73.52 (±16.36)	72.35 (±16.03)	74.67 (±18.23)	75.21 (±17.65)	75.38 (±18.23)	70.00 (±11.66)	
**Height (m)**							<0.001 *^a^
Mean (±SD)	1.68 (±0.0784)	1.63 (±0.084)	1.68 (±0.070)	1.69 (±0.067)	1.69 (±0.087)	1.70 (±0.088)	
**BMI (kg/m^2^)**							0.144 ^a^
Mean (±SD)	26.00(±5.22)	27.18 (±5.63)	26.27 (±5.67)	26.15 (±5.57)	26.18 (±5.43)	24.22 (±3.80)	

^a^ One-way ANOVA—excluding the overall data. ^b^ Fisher–Freeman–Halton test. * Significant *p* < 0.05.

**Table 2 ijerph-20-04443-t002:** Description and comparison between all product users.

Variables	CCs (n = 152)	ECs (n = 66)	HTPs (n = 11)	Poly-Users (n = 86)			*p*-Value
				CCs (n = 86)	ECs (n = 86)	HTPs (n = 7)	
**Age start (years)**							<0.001 *^a^ <0.001 *^b^
Mean (±SD)	17.41 (±3.84)	26.65 (±8.09)	35.00 (±7.47)	17.16 (±3.66)	27.20 (±8.29)	27.00 (±9.79)	
Range	(10–31)	(13–47)	(22–46)	(10–29)	(15–52)	(18–48)	
≤18	110 (72.4)	11 (16.7)	0 (0.0)	56 (65.1)	8 (9.3)	1 (14.3)	
>18	42 (27.6)	55 (83.3)	11 (100.0)	30 (34.9)	78 (90.7)	6 (85.7)	
**Duration smoking (years)**							<0.001 *^a^ <0.001 *^b^
Mean (±SD)	17.58 (±9.74)	4.20 (±3.61)	4.32 (±1.76)	13.93 (±7.76)	4.20 (±3.13)	3.43 (±1.51)	
Range	(2–45)	(1–22)	(1–8)	(1–36)	(1–19)	(2–6)	
0–5	17 (11.2)	51 (77.3)	9 (81.8)	13 (15.1)	66 (76.7)	6 (85.7)	
≥5	135 (88.8)	15 (22.7)	2 (18.2)	73 (84.9)	20 (23.3)	1 (14.3)	
**Cost/month (MYR)**							<0.001 *^a^ <0.001 *^b^
Mean (±SD)	197.61 (±139.98)	81.86 (±96.30)	330.64 (±195.56)	187.66 (±128.53)	82.49 (±97.86)	152.14 (±122.13)	
Range	(10–700)	(7.5–700)	(28–630)	(15–600)	(15–600)	(15–300)	
≤100	50 (32.9)	54 (81.8)	2 (18.2)	32 (37.2)	70 (81.4)	3 (42.9)	
>100	102 (67.1)	12 (18.2)	9 (81.8)	54 (62.8)	16 (18.6)	4 (57.1)	
**Fagerström score**							0.109 ^a^ 0.195 ^b^
Mean (±SD)	3.66 (±2.19)	4.12 (±2.39)	2.55 (±2.21)	3.38 (±2.27)	3.81 (±2.20)	4.00 (±1.41)	
Range	(0–9)	(0–10)	(0–6)	(0–9)	(0–9)	(1–5)	
<4	73 (48.0)	29 (43.9)	7 (63.6)	41 (47.7)	45 (52.3)	1 (14.3)	
4–6	62 (40.8)	25 (37.9)	4 (36.4)	38 (44.2)	29 (33.7)	6 (85.7)	
7–10	17 (11.2)	12 (18.2)	0 (0)	7 (8.1)	12 (14.0)	0 (0)	
**Attempt to quit**							<0.001 *^b^
Yes	85 (55.9)	23 (34.8)	1 (9.1)	74 (86.0)	41 (47.7)	2 (28.6)	
No	67 (44.1)	43 (65.2)	10 (90.9)	12 (14.0)	45 (52.3)	5 (71.4)	

^a^ One-way ANOVA. ^b^ Pearson Chi-square. * Significant *p* < 0.05.

**Table 3 ijerph-20-04443-t003:** Comparison of exhaled CO level according to different group of products used and current non-smokers.

Smoking Status	Mean Duration Use (Years)	Minimum–Maximum (ppm)	Median (IQR) (ppm)	*p*-Value
				<0.001 ^a^
**CCs only (n = 152)**	17.58 (±9.74)	2–50	13.00 (9.00–19.75)	
**PUs (n = 86)**	13.93(±7.76) ^b^, 4.20 (±3.13) ^c^, and 3.43(±1.51) ^d^	1–39	7.00 (3.00–12.25)	
**ECs only (n = 66)**	4.20 (±3.61)	1–12	2.00 (1.00–3.00)	
**HTPs only (n = 11)**	4.32 (±1.76)	1–3	2.00 (1.00–3.00)	
**Non-smokers (n = 342)**	Not applicable	0–7	1.00 (1.00–2.00)	

^a^ Kruskal–Wallis H test, ^b^ CC (PU group), ^c^ EC (PU group), and ^d^ HTP (PU group).

**Table 4 ijerph-20-04443-t004:** Exhaled CO level categories according to users and non-smokers.

Smoking Status	Mean Duration Use (Years)	eCO Level (ppm)			*p*-Value
		0–6 n (%) (n = 481)	7–10 n (%) (n = 50)	>10 n (%) (n = 126)	
					<0.001 ^a^
Non-smokers (n = 342)	Not applicable	337 (98.5)	5 (1.5)	0 (0)	
CCs only (n = 152)	17.58 (±9.74)	28 (18.4)	23 (15.1)	101 (66.4)	
PUs (n = 86)	13.93(±7.76) ^b^, 4.20 (±3.13) ^c^, and 3.43 (±1.51) ^d^	42 (48.8)	20 (23.3)	24 (27.9)	
ECs only (n = 66)	4.20 (±3.61)	63 (95.5)	2 (3.0)	1 (1.5)	
HTPs only (n = 11)	4.32 (±1.76)	11 (100.0)	0 (0)	0 (0)	

^a^ Pearson Chi-square test. ^b^ CC (PU group). ^c^ EC (PU group). ^d^ HTP (PU group).

## Data Availability

Not applicable.

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
