# Peer review of "Exhaled Carbon Monoxide Level and Practices among Tobacco and Nicotine Adult Users in Klang Valley, Malaysia"

_ijerph, 2023, doi:10.3390/ijerph20054443_

Round 1

Reviewer 1 Report

The manuscript “Exhaled Carbon Monoxide Level and Practices Among Tobacco 2 and Nicotine Adult Users in Klang Valley, Malaysia” adds to the small but growing literature on heated tobacco products.  Overall, the authors report some differences in demographics and tobacco use behavior among different tobacco user groups.  Not surprisingly, exhaled CO was lower in exclusive users of e-cigarettes and heated tobacco products compared with exclusive cigarette smokers.  Below are a few suggestions for strengthening the paper.

Methods/Results

Table 1 essentially presents the same variables as Table 2 but for the overall population vs. by tobacco group.  Consider combining Table 1 and Table 2 by adding an overall column.

Tables 2 and 3:  The row percents make it difficult to assess differences among the tobacco groups.  Consider present column percents rather than row percents.

Table 3:  Missing from the table is information on frequency and intensity of use of each tobacco product.  If collected, consider presenting percentage of everyday use of each product, intensity of use (eg number of ecigs/heatsticks per day; cigarettes per day), flavored tobacco product use, marijuana use.

Tables 4-5: As mentioned above, it’s difficult to tell if differences in exhaled CO levels are due to differences in the products themselves or differences in intensity/frequency of use of each product.  Consider stratifying mean exhaled CO by everyday vs someday use or high vs. low intensity use of each product. 

Limitations:  I suggest adding a limitation on the limited sample size, especially for heated tobacco product use.

Minor comments

I suggest having a copyeditor review the manuscript to correct typos and for word flow.  Some suggestions below:

Lines 61-62: “It was reported that an estimated that in 2000, 32.7% of the global population aged 15 years and older were current users of some  form of tobacco and in 2020, the prevalence declined one third to 22.3%.”

Suggested edit: “It was reported that in 2000, an estimated 32.7% of the global population aged 15 years and older were current users of some form of tobacco and in 2020, the prevalence declined one third to 22.3%.”

Line 187 “Participants were only required to answer sections of questionnaire based on their current use.”

Suggested edit: “Participants were only required to answer sections of the questionnaire based on their current use.”

Line 534 “Exhaled CO levels between the different groups 533 showed significant differences and consistent with other available studies.”

Suggested edit: “Exhaled CO levels between the different groups 533 showed significant differences and is consistent with other available studies.”

Reviewer 2 Report

In the submitted paper, the authors discuss an important topic: the problem of cigarette smoking in its various aspects and forms. 

In my opinion, ''The Introduction” part is overextended and too long – the authors present information on the epidemiology of cigarette smoking, its socioeconomic aspects, various devices used to deliver nicotine (including conventional cigarettes (CC), heated tobacco products (HTP) and electronic cigarette (EC)), and the possibility of using nicotine replacement therapy to treat nicotine addiction which is not reflected in further parts of the work.

This study aims to determine the practices, nicotine dependency profile, association with exhaled carbon monoxide level, and pulmonary function among adult product users and non-smokers. The methodology used a questionnaire created for the purpose of the study, taking into account the past, and current forms of smoking and the Fagenstrōm nicotine addiction test, as well as measurements of carbon monoxide (CO) concentration in exhaled air. 

In the third part, the obtained results are presented in the form of tables and graphs, taking into account the age at which the product was used, the duration of its use, the current smoking status among participants, a comparison of practices, and a profile of nicotine addiction, and patterns of "switching" smoking single, double or multi-user. It was found that the obtained results of CO concentration measurements depend on the methods of smoking and burning tobacco, and the construction of devices.

The discussion refers to the results obtained in the individual analyzed categories and their confrontation with the results of other authors, both national and international. Unfortunately, I cannot agree with the last conclusion in section 5, supported by the previous reference to number 38. In my opinion, any form of smoking using various devices, such as EC, HTP, and ENDS, has a detrimental effect on health in general and on the lungs in particular, so should not be alternative methods to CC.

In the opinion of the reviewer, the work can be qualified as a publication in the ''International Journal of Environmental Research and Public Health” after the correction of  ''The Introduction” part. 

Reviewer 3 Report

This study focuses on the epidemiology of tobacco use in the Klang Valley region of Malaysia. Due to the thousands of harmful chemicals that smokers and passive smokers inhale, various diseases, such as lung cancer, can wreak havoc on the lungs. The tobacco use age is declining due to the variety and availability of tobacco products. The use of small regional population surveys may help understand the distribution of smokers and their smoking habits and patterns. Therefore, I would like to ask some questions about this research:

1. Using questionnaires, respondents tend to have avoidance issues, especially among young people. Although the authors used the picO™ Smokerlyzer® to detect tobacco product use among respondents, the sensitivity and accuracy of the eCO test were not superior to that of cotinine. The use of the NicAlertR assay in saliva to verify smoking status is well documented (J Clin Diagn Res. 2016 Mar; 10(3): ZE04–ZE06.). It is recommended to use both methods for detection.

2. Tables 2 and 3 were statistically analyzed using one-way ANOVA. However, the tobacco use formula variable is not continuous, so this statistical approach is not applicable. Please use an appropriate statistical method or rearrange the table.

3. The expression in Figure 2 is not easy to understand. Is it more appropriate to use a flowchart to present it?

4. Please explain why age 30 in Table 2 is the cut-off point. In addition, the number of tobacco users in Table 2 and Table 3 is inconsistent. Please confirm again.
